# Quantifying the Response of German Forests to Drought Events via Satellite Imagery

Marius Philipp [1,2,*], Martin Wegmann [1] and Carina Kübert-Flock [1,3]

1   Department of Remote Sensing, Institute of Geography and Geology, University of Wuerzburg,
    D-97074 Wuerzburg, Germany; martin.wegmann@uni-wuerzburg.de (M.W.);
    carina.kuebert-flock@hlnug.hessen.de (C.K.-F.)
2   German Remote Sensing Data Center (DFD), German Aerospace Center (DLR), Muenchner Strasse 20,
    D-82234 Wessling, Germany
3   Hessian Agency for Nature Conservation, Environment and Geology, Rheingaustr. 186,
    D-65203 Wiesbaden, Germany
*   Correspondence: marius.philipp@uni-wuerzburg.de

**Abstract:** Forest systems provide crucial ecosystem functions to our environment, such as balancing carbon stocks and influencing the local, regional and global climate. A trend towards an increasing frequency of climate change induced extreme weather events, including drought, is hereby a major challenge for forest management. Within this context, the application of remote sensing data provides a powerful means for fast, operational and inexpensive investigations over large spatial scales and time. This study was dedicated to explore the potential of satellite data in combination with harmonic analyses for quantifying the vegetation response to drought events in German forests. The harmonic modelling method was compared with a z-score standardization approach and correlated against both, meteorological and topographical data. Optical satellite imagery from Landsat and the Moderate Resolution Imaging Spectroradiometer (MODIS) was used in combination with three commonly applied vegetation indices. Highest correlation scores based on the harmonic modelling technique were computed for the 6th harmonic degree. MODIS imagery in combination with the Normalized Difference Vegetation Index (NDVI) generated hereby best results for measuring spectral response to drought conditions. Strongest correlation between remote sensing data and meteorological measures were observed for soil moisture and the self-calibrated Palmer Drought Severity Index (scPDSI). Furthermore, forests regions over sandy soils with pine as the dominant tree type were identified to be particularly vulnerable to drought. In addition, topographical analyses suggested mitigated drought affects along hill slopes. While the proposed approaches provide valuable information about vegetation dynamics as a response to meteorological weather conditions, standardized in-situ measurements over larger spatial scales and related to drought quantification are required for further in-depth quality assessment of the used methods and data.

**Keywords:** time-series; harmonic analysis; z-score; scPDSI; drought; vegetation response; forest ecosystems; Google Earth Engine

## 1. Introduction

An ongoing global climate warming process has been accepted by scientists worldwide for years. Influences of climate change can be perceived everywhere in our environment, especially in the phenology of plants [1]. Forests are particularly sensitive to climate change due to their relatively long lifespan and limited adaptation skills to altering environmental conditions [2]. In 2018, the total quantity of damaged wood in Germany reached almost 32 million solid cubic meters and further increased to roughly 46 million solid cubic meters in 2019 [3]. The analysis of ecological and economic effects as well as the development of silvicultural measures, which lead to resistant and resilient forest ecosystems in the face of climate change, are hereby identified as one of the biggest challenges [4].

Forests directly affect local ecosystems, the global climate and, as carbon stores, balance global carbon stocks [5]. Therefore, the health status of trees is of fundamental interest to national and international sustainable forest management. The biggest influencing factors are thereby the damage caused by air pollution and the effects of climate change, including drought [6]. Despite their high relevance, the effects of drought on forest ecosystems are still not fully understood [7]. Due to logistical and financial limitations, large-scale forest conversion measures are not practically feasible in the short term which in turn suggests that the focus must be placed on those regions that are already characterized by warm and dry climatic conditions [8]. In a study by Reif et al. [4], numerous forestry and nature conservation experts were interviewed to name regions in Germany, which they believe are most affected by climate change. Regions with pine and spruce stands were hereby described as heavily endangered, especially spruces on planar to montane sites as well as locations with poor water supply. Generally, an increased vulnerability in coniferous-dominated stands is expected [4]. In addition, spruce as host trees are affected by bark beetle infestation to a large extent and, as a shallow-rooted tree species, are also highly vulnerable to storms and droughts [9,10]. This has a strong impact on the spruce wood prices which have fallen sharply due to the oversupply of timber harvesting [11–13]. Pine stands are also severely troubled by the current climatic conditions. After the hot and dry summer of 2018, about half of the pine trees that were located in the Karlsruher Hardtwald and already weakened by drought stress, died [14]. Drought events further promote the vulnerability of trees and therefore increase the risk of forest fires and pest infestations by e.g., the bark beetle [15–19]. Current trends in climate change suggest a higher frequency of droughts as well as other associated disturbing factors such as wind throws, forest fires and pest infestations in future forests [20].

In order to study the impact of droughts on forest systems, many approaches were already applied using in-situ data. Yet this type of study is usually heavily limited by its area coverage and, in addition, time-consuming and labor-intensive. Furthermore, numerous indicators from in-situ studies are based on expert opinions [21]. However, these current challenges require a near-real-time and operationally applicable analysis of forest ecosystems, especially in terms of variation over space and time. In this context, remote sensing offers the opportunity to investigate the condition of forests with a large area coverage in a comparable, reproducible and cost-effective manner [6]. Freely available tools e.g., the cloud-based geospatial analysis platform Google Earth Engine (GEE) make it easier than it has ever been before to work with mass data and long time-series [22]. Remote sensing data has been successfully applied in numerous studies to monitor and analyze forests at both global and regional scale [23–27]. In a study by Bochenek et al. [28], high-resolution satellite imagery was used in combination with various vegetation indices in order to study the effects of changing climatic conditions in selected Polish forest regions. In particular, indices related to water stress enabled an analysis of drought events within the area of study [28]. In another work by Lewińska et al. [29], drought events in alpine forests in South Tyrol were investigated. A synergistic use of meteorological-based indices together with satellite-based vegetation indices was tested in their study. The authors identified a prolonged drought condition between 2003 and 2007 as well as a general drying tendency [29]. As mentioned in a recent review about remote sensing in German forests by Holzwarth et al. [30], monitoring forest areas via remote sensing gained increasing interest over the last two decades. However, the authors also stress the lack of studies on a national scale in contrast to local investigations [30].

Towards the goal of fully exploiting the monitoring potential of the satellite systems, challenges such as noise and data gaps must be effectively addressed. These quality losses are mainly caused by sensor artifacts, clouds, cloud shadows and other weather conditions [31]. In the context of time-series analyses, harmonic modeling is a powerful tool to fill these gaps and reduce noise by smoothing the original signal [32]. Various studies have demonstrated the successful implementation of harmonic analyses for monitoring the Earth's surface and improving classification results [33–36]. Next to improving clas-

sification results, this approach has also been used to identify changes in agricultural areas [37], landscape fragmentation [38] as well as investigating changes in land cover classes [39]. The application of the harmonic modeling generated hereby more accurate results compared to other established methods. Despite various successful implementations of this promising approach, so far, very little attention was given to the integration of harmonic computations in the context of drought-related vegetation response within forest ecosystems. Tools such as the "rHarmonics" R-package, which allows for easy computation of harmonic curves on time-series data, will hereby enable a wider audience to apply this methodology in future studies [40].

This study aims to close existing gaps in the current state of research in order to enable better spatio-temporal planning for forest ecosystems in the context of climate change-induced extreme weather events. The main focus lies on the identification and quantification of vegetation response to meteorological drought conditions. Meteorological drought in Germany was analyzed using publicly available weather data provided by the Deutscher Wetterdienst (DWD). Optical remote sensing imagery from the Moderate Resolution Imaging Spectroradiometer (MODIS) and the Landsat satellite missions were utilized for vegetation response assessment. Three indices (Normalized Difference Vegetation Index (NDVI), Normalized Burn Ratio (NBR), Normalized Difference Moisture Index (NDMI)) and two methods (harmonic modeling and z-score analysis), were tested together with the satellite data for the identification of the best performing predictor combination. Furthermore, different tree species were compared in their susceptibility to drought with respect to the present soil types and topographic positioning.

## 2. Material and Methods

Investigations on drought vulnerability were conducted on two levels (Figure 1). The first level (Analysis Level 1) was designed to identify the best combination of remote sensing data, vegetation index and harmonic degree for analyzing the vegetation response to drought events on a national scale. The purpose of the second level (Analysis Level 2) was to conduct detailed analyses on different tree species in their susceptibility to such extreme events on local scales by using the previously identified best predictor combination. The influence of soil type and the topographic parameters slope and aspect were also considered. Details on the used data, the pre-processing steps, the implemented algorithms for quantifying vegetation response, and the applied analysis on national (Analysis Level 1) and further local scales (Analysis Level 2) as visualized in Figure 1 are provided from here onwards.

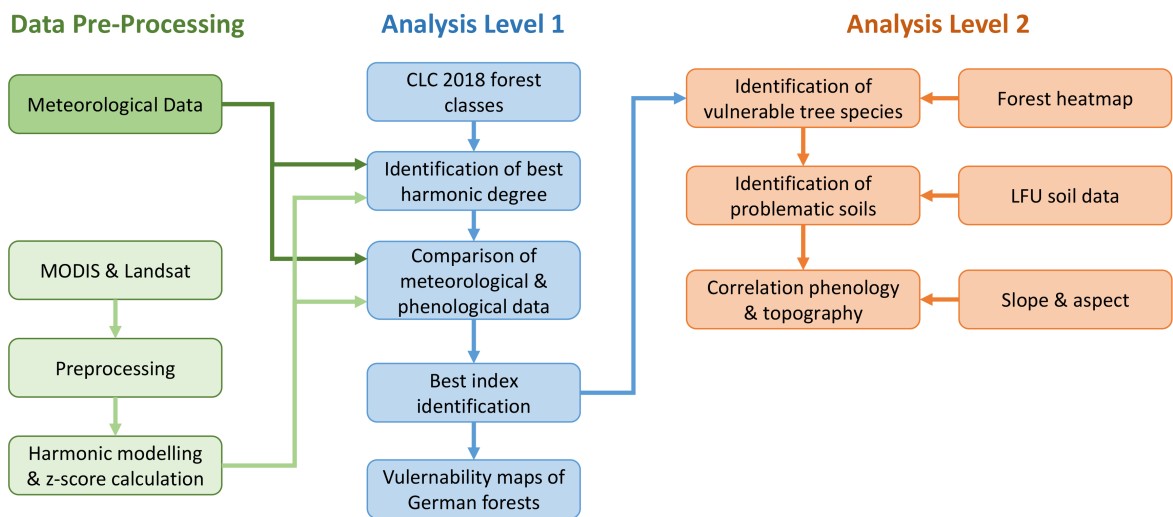

**Figure 1.** Flowchart of the study outline. The analysis was conducted on two levels. The first level was designed to identify optimal predictor combinations for drought assessment on a national scale. The second level was dedicated to study local drought response of different tree types depending on present soil types and topographic positioning.

*2.1. Materials*

The following chapter provides a description of all materials used throughout this study. Detailed information on data type, spatio-temporal resolution and temporal range as well as the associated Analysis Level is also listed in Table 1.

**Table 1.** Name, spatio-temporal resolution and temporal range of applied data and the associated analysis level during this study.

| Name | Spatial Resolution | Temporal Resolution | Temporal Range | Analysis Level | Ref. |
|---|---|---|---|---|---|
| **Satellite Data** | | | | | |
| MODIS Terra (product MOD09A1) | 500 m | 8 days (aggregated to monthly medians) | since 2000 (subsetted to 2000–2019) | Analysis Level 1 | [41] |
| Landsat-4, -5, -7, -8 (surface reflectance products) | 30 m (resampled to 60 m) | 16 days (aggregated to monthly medians) | since 1982 (focus on August 2018) | Analysis Level 2 | [42,43] |
| **Meteorological Data** | | | | | |
| Precipitation | 1 km | monthly | since 1881 (subsetted to match MODIS) | Analysis Level 1 | [44] |
| Potential evapotranspiration | 1 km | monthly | since 1991 (subsetted to match MODIS) | Analysis Level 1 | [44] |
| Maximum air temperature | 1 km | monthly | since 1901 (subsetted to match MODIS) | Analysis Level 1 | [44] |
| **Soil Data** | | | | | |
| Soil moisture | 1 km | monthly | since 1991 (subsetted to match MODIS) | Analysis Level 1 | [44] |
| Soil types | scale of 1:25,000 | unitemporal | 2019 | Analysis Level 2 | [45] |
| **Topographic Data** | | | | | |
| Slope and aspect derived from a Digital Terrain Model (DTM) | 25 m (resampled to Landsat data) | unitemporal | 2019 | Analysis Level 2 | [46] |
| **Forest Data** | | | | | |
| Corine Land Cover (CLC) | 100 m | unitemporal | 2018 | Analysis Level 1 | [47] |
| Forest heatmap (60 forest reference areas in Bavaria) | scale of 1:10,000 | unitemporal | 2003–2018 | Analysis Level 2 | [48] |

2.1.1. Forest Data

To avoid biased results due to local environmental conditions, analyses on the relationship between meteorological and vegetational information (Analysis Level 1) were conducted on a national scale across all German forest regions. The publicly available Corine Land Cover (CLC) 2018 raster data set was used as a reference which provides land cover information with a spatial resolution of 100 m [47]. The forest classes 311 (broad-leaved forest, ~35,920 km$^2$), 312 (coniferous forest, ~58,644 km$^2$) and 313 (mixed forest, ~13,566 km$^2$) were used to create a German-wide forest mask (Figure 2a) [47]. It has to be noted, that this study assumes no land cover change, e.g., in terms of deforestation or agricultural expansion within the CLC 2018 forest area over the investigated study time period.

Within the framework of this study, the Bayerische Staatsforsten (BaySF) provided a reference shapefile that covers 60 forest areas (~229 km$^2$) all over Bavaria which were used for the second analysis level (Figure 2b). Each forest reference area shows the local dominant tree species as well as forest areas that are older than 80 years. This data set was also utilized to study the influence of different soil types and topographic parameters on the drought susceptibility of forest sites.

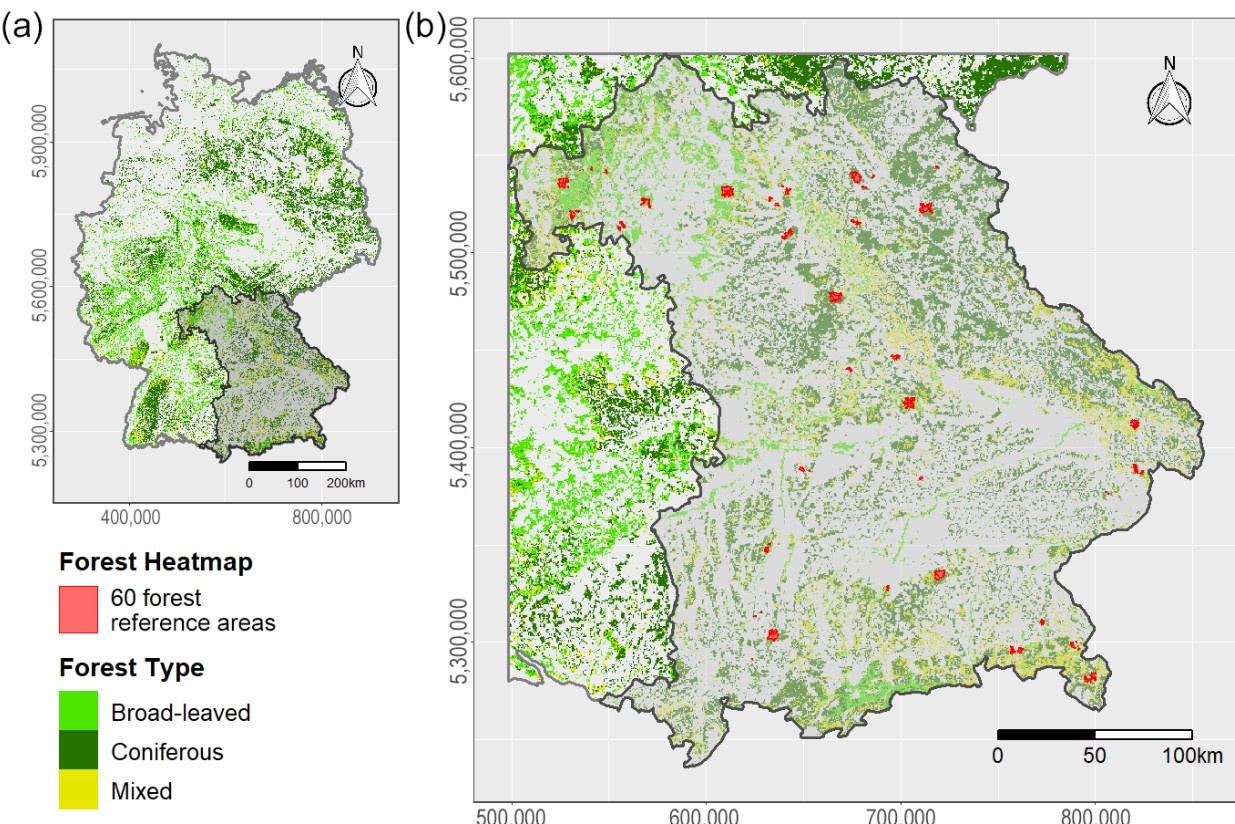

**Figure 2.** Study area covering all German forest regions for Analysis Level 1 (**a**), as well as 60 forest reference areas across Bavaria with information about dominant tree species and forest age for Analysis Level 2 (**b**).

### 2.1.2. Meteorological Data

Monthly precipitation, potential evapotranspiration and maximum air temperature data with a spatial resolution of 1 km was downloaded as ASCII grids via the DWD Climate Data Center (CDC) [44]. Additionally to the mentioned meteorological data sets, soil moisture measurements provided by the DWD as ASCII grids with the same spatial resolution of 1 km were downloaded. The gridded values are based on weather station data which have been horizontally interpolated via an inverse distance weighting (IDW) approach for grid cells with no available station data [49].

### 2.1.3. Soil Type Data

Since the status of a forest also strongly depends on the present soil and nutrient composition [50,51], different soil types in their contribution to the drought vulnerability of forests were compared. Soil data was obtained via the Bavarian Landesamt für Umwelt (LFU) which provides shapefiles with soil cover information of Bavaria on a scale of 1:25.000. The data is conceptually derived from existing documents and was both evaluated and supplemented by field site inspections [45].

### 2.1.4. Topographic Data

A Digital Terrain Model (DTM) with a spatial resolution of 25 m was used as a basis for the topographic calculations. The DTM was obtained via the Bundesamt für Kartographie und Geodäsie (BKG) and was generated through different methods, including laser scanning, photogrammetry and digitization of contour lines [46].

### 2.1.5. Remote Sensing Data

Remote sensing data from the Landsat satellite mission and MODIS onboard the Terra satellite was utilized in this study. For Landsat data, atmospherically corrected Tier 1

Surface Reflectance (SR) imagery from the satellites Landsat-4, -5, -7 and -8 were used and combined. SR data provides atmospherically corrected imagery which accounts for noise effects such as thin clouds or aerosol scattering [42,43]. This results in better comparability between different scenes that cover the same region [52]. Thus, continuous optical remote sensing data since 1982 with a spatial resolution of 30 m was generated. The resolution of each Landsat scene was subsequently lowered by a factor of 2 in order to reduce the potential negative influence of pixel geolocation error uncertainties [53–55]. Therefore, Landsat data with a 60 m resolution was generated and used for drought response analyses.

Complementary to Landsat imagery, the MODIS Terra MOD09A1 Version 6 product was included which provides atmospherically corrected SR data with a spatial resolution of 500 m and a temporal resolution of eight days [41]. Each pixel hereby represents the best possible observation within an eight-day time period [56]. By including both Landsat and MODIS imagery, the influence of spatial and temporal resolution for detecting the vegetation response could be compared.

*2.2. Data Preprocessing*

2.2.1. Calculation of the Self-Calibrated Palmer Drought Severity Index (scPDSI)

Using the previously described meteorological data, the self-calibrated Palmer Drought Severity Index (scPDSI), which is based on the Palmer Drought Severity Index (PDSI) by Palmer [57], was derived as a meteorological drought reference. The PDSI is one of the most widely used drought indices to quantify long-term drought conditions. Instead of precipitation anomalies, the concept of the PDSI is based on a water supply-and-demand approach that makes it more comprehensive than precipitation-only indices [58]. A full description of the framework for this model can be found in Palmer [57], but we provide a brief overview of the procedure for estimating the PDSI here.

For each month of the study period the four variables recharge ($R$), runoff ($RO$), evapotranspiration ($ET$) and loss ($L$) as well as their associated potential values potential recharge ($PR$), potential runoff ($PRO$), potential evapotranspiration ($PET$) and potential loss ($PL$) are calculated [57]. In order to provide the appropriate existing climatic conditions (EEC), the four potential variables are weighted based on the water balance coefficients ($\alpha$, $\beta$ $\gamma$, $\sigma$) which can be estimated for a given month $i$ in the following manner [59]:

$$\alpha_i = \frac{\overline{ET_i}}{\overline{PE_i}} \qquad \beta_i = \frac{\overline{R_i}}{\overline{PR_i}} \qquad \gamma_i = \frac{\overline{RO_i}}{\overline{PRO_i}} \qquad \sigma_i = \frac{\overline{L_i}}{\overline{PL_i}} \tag{1}$$

The bar over a term specifies hereby an average value. The combination of the EEC potential values yields the amount of precipitation $\hat{P}$ that is needed to maintain a normal soil moisture level for a given month [57]. Subtracting $\hat{P}$ from the actual amount of precipitation of the respective month forms the moisture departure $d$ [59]:

$$d = P - \hat{P} = P - (\alpha_i PE + \beta_i PR + \gamma_i PRO + \sigma_i PL). \tag{2}$$

In order to further adjust $d$ to different climatic conditions of a given area, Palmer [57] derived $K$ which represent the local climatic characteristics. $\overline{D_i}$ is hereby the average moisture departure of a given month [59]:

$$K'_i = 1.5 \log_{10}\left(\frac{\frac{\overline{PE_i} + \overline{R_i} + \overline{RO_i}}{\overline{P_i} + \overline{L_i}} + 2.8}{\overline{D_i}}\right) + 0.5 \tag{3}$$

$$K_i = \frac{17.67}{\sum_{j=1}^{12} \overline{D_j} K'_j} K'_i. \tag{4}$$

The empirical constant with the value 17.67 in Equation (4) was derived by using data from nine locations across United States [57]. Multiplying the moisture departure $d$ by $K$ results in the anomaly index $Z$ (Equation (5)) which in turn describes the wetness

or dryness of a given month without the consideration to recent precipitation trends [59]. Ultimately, the PDSI can be calculated using the general Equation (6) [57]:

$$Z = dK \tag{5}$$

$$X_i = 0.897X_{i-1} + (\frac{1}{3})Z_i. \tag{6}$$

However, the model is being criticized for using empirical constants that were derived through a limited amount of samples and locations which makes a spatial comparison of PDSI values complicated, especially across different climatic regions [60]. Wells et al. [59] addressed this flaw for the scPDSI by replacing the empirical constants, in particular the empirically derived climatic characteristics $K$ (Equation (4)) as well as the duration factors 0.897 and $\frac{1}{3}$ (Equation (6)), with values that are dynamically calculated from the input data at a given location. The algorithm can be applied on continuous monthly weather data with a temporal coverage of at least 25 years [59]. The index was chosen as a meteorological drought reference since several studies demonstrated its good correlation to the status of forest ecosystems [29,61–63]. In order to calculate the scPDSI, the R package of the same name scPDSI v. 0.1.3 was applied which uses both precipitations as well as potential evapotranspiration as input data [64]. While precipitation data is available since the year 1881, data for the potential evapotranspiration is only available since 1991. Therefore, monthly scPDSI values were calculated for every pixel from January 1991 until December 2019. Table 2 categorizes the scPDSI values into different drought classes after Wells et al. [59]. Appendix A Figure A1 demonstrates an exemplary mean scPDSI time-series of the Steigerwald, a forest in northern Bavaria.

**Table 2.** Drought classification after Wells et al. [59].

| scPDSI Values | scPDSI Category |
| --- | --- |
| Above 4.00 | Extreme wet spell |
| 3.00 to 3.99 | Severe wet spell |
| 2.00 to 2.99 | Moderate wet spell |
| 1.00 to 1.99 | Mild wet spell |
| 0.50 to 0.99 | Incipient wet spell |
| 0.49 to −0.49 | Normal |
| −0.50 to −0.99 | Incipient drought |
| −1.00 to −1.99 | Mild drought |
| −2.00 to −2.99 | Moderate drought |
| −3.00 to −3.99 | Severe drought |
| Below −4.00 | Extreme drought |

2.2.2. Cloud Masking and Calculation of Monthly Median Satellite Images

In order to reduce the effects of cloud contamination, the "pixel_qa" band of the Landsat SR product was used which provides a cloud, cloud shadow and snow mask based on the CFMask algorithm [43]. Similarly, the "StateQA" layer of the MOD09A1 product was employed for removing clouds, cloud shadows and snow from the MODIS data [56].

For better data comparability, monthly median images were computed for both Landsat and MODIS data. Since the aim of this study is to analyze changes in the spectral signal of vegetation as a response to extreme weather events, data values have to be as representative as possible in order to detect actual change instead of noise. Creating a median image for each month further reduces any noise leftovers within the data sets. The median, as a statistical parameter, has the advantage of being robust to outliers [65]. Furthermore, it facilitates comparability since a consistent temporal spacing between the values of all data sets is being realized. If only one scene was available for a given month, no median was computed but the single observation was used as a representation for that month. Moreover, any correlation analysis conducted in this study was only applied for

months with available data. Therefore, in case there was no available data after cloud, cloud shadow and snow masking for a given month, said month was skipped during the correlation testing.

2.2.3. Calculation of Vegetation Indices

The covered wavelengths of individual bands from different sensors are listed in Table 3 and visualized in Appendix A Figure A2. Three different vegetation indices were applied and compared in their ability to detect drought-related vegetation response.

**Table 3.** Spectral wavelengths covered by different sensors. Bands of the sensors Thematic Mapper (TM) [66], Enhanced Thematic Mapper Plus (ETM+) [67], Operational Land Imager (OLI) [67] and the Moderate Resolution Imaging Spectroradiometer (MODIS) product MOD09A1 [56] are compared. Bands in the visible Red-Green-Blue (RGB), Near Infrared (NIR) and Short Wavelength Infrared (SWIR) wavelength areas are listed.

| Bands | TM | ETM+ | OLI | MOD09A1 |
|-------|----|------|-----|---------|
| Blue | 450–520 nm | 450–515 nm | 450–515 nm | 459–479 nm |
| Green | 520–600 nm | 525–605 nm | 525–600 nm | 545–565 nm |
| Red | 630–690 nm | 630–690 nm | 630–680 nm | 620–670 nm |
| NIR1 | 760–900 nm | 775–900 nm | 845–885 nm | 841–876 nm |
| NIR2 | – | – | – | 1230–1250 nm |
| SWIR1 | 1550–1750 nm | 1550–1750 nm | 1560–1660 nm | 1628–1652 nm |
| SWIR2 | 2080–2350 nm | 2090–2350 nm | 2100–2300 nm | 2105–2155 nm |

The first index is the Normalized Difference Vegetation Index (NDVI) which is one of the most widely used vegetation indices. It was applied in numerous studies for vegetation health condition or greenness detection [68–70]. The NDVI is calculated using the red and Near Infrared (NIR) 1 band (Formula (7)) [71].

$$NDVI = \frac{NIR1 - RED}{NIR1 + RED}. \tag{7}$$

The second index in this study is the Normalized Burn Ratio (NBR) that includes the red and Short Wavelength Infrared (SWIR) 2 band (Formula (8)) [72]. The index has been frequently used to study fire severity and for mapping burnt areas [73–76], but also for forest degradation and recovery analysis [77–79].

$$NBR = \frac{NIR1 - SWIR2}{NIR1 + SWIR2} \tag{8}$$

The last index included in this work is the Normalized Difference Moisture Index (NDMI) (also called Normalized Difference Water Index (NDWI)) by Gao [80] and utilizes the NIR 1 and SWIR 1 bands (Formula (9)). The index was also applied in several forest-related studies [81–83].

$$NDMI = \frac{NIR1 - SWIR1}{NIR1 + SWIR1}. \tag{9}$$

*2.3. Capturing the Vegetation Response to Drought Events*

Harmonic analyses were utilized for the identification of vegetation response to drought events. Complementary to the harmonic modeling, the well established z-score algorithm was applied as a second drought estimator based on the spectral behavior of vegetation which demonstrated its high potential as a drought and forest health index in several studies [29,84,85].

Before applying any vegetation response analysis, the time-series data sets were detrended using a simple linear Ordinary Least Squares (OLS) regression approach that resulted in stationary data. Therefore, seasonal vegetational changes could be analyzed without the influence of a background trend [86,87]. Harmonic modeling and z-score

computation were performed using GEE, since it provides the computational means necessary for carrying out this kind of analysis on a national scale. The GEE scripts for z-score computation and harmonic modeling are available on GitHub (https://github.com/MBalthasar/Harmonics_and_z-score_GEE_code) and may serve as a valuable tool for similar studies.

### 2.3.1. Harmonic Modelling

To calculate the harmonic fitted curve of a periodic signal, OLS regressions were computed using coupled sine and cosine curves on time-series data [33,88,89]. The first step was to calculate the time-difference between each time step and a reference date. Since GEE stores time information in units of unix time, which is the elapsed time since midnight 1 January 1970 in milliseconds [90], said the date was chosen as the reference date. The resulting difference values were subsequently converted to the temporal difference in fractal years $t$, followed by a conversion into radians.

Depending on the number of cycles $n$ that should be modeled per year, sine and cosine values of 1:$n$ * radians were calculated. Afterwards, a multiple linear OLS regression calculation was performed with the respective vegetation index values as the dependent variable. The radians, sine and cosine values as well as a constant of 1 served as the independent variables. The regression analysis outputs a coefficient value $\beta$ for each independent variable. The coefficient $\beta_0$ for the added constant represents hereby the intercept value. After multiplying the independent variables with their respective coefficients based on the OLS regression, the sum of all independent values was calculated which resulted in the fitted value $X_t$. The underlying algorithm for the harmonic analysis is shown in Formula (10).

As previously mentioned, different numbers of cycles per year can be modeled. The number of modeled cycles hereby defines the degree of the harmonic analysis. On the one hand, increasing the number of periodic signals per year can lead to more representative curves, especially if the original signal does not represent a perfect sinusoidal curve [91]. On the other hand, higher harmonic degrees tend to generate erroneous spikes and oscillations [92].

In order to counteract this problem, the maximum harmonic degree was defined after the Nyquist-Shannon theorem. Said theorem originally addressed the discretization of analog signals and states that in order to reconstruct a signal, the sampling rate of the original signal has to be at least twice as high as its component with the highest frequency [93,94]. As mentioned by Ficker and Martišek [95], this minimum sampling rate is hereby commonly referred to as the Nyquist rate.

Since time-series data in this study is based on monthly values, the sampling rate of the original signal equals twelve. With respect to the mentioned Nyquist rate, a maximum of six cycles per year can be modeled using this sampling strategy. In addition to the 6th degree harmonic model, a 1st and 3rd degree harmonic modeling approach was performed in order to compare and identify the harmonic degree that produces the highest quality results. A 3rd degree harmonic fitted curve refers hereby to the sum of 1–3 modeled cycles per year. A sixth degree harmonic model refers to the sum of 1–6 modeled cycles per year, respectively.

Having modeled the harmonic fitted curve, it was interpreted as the ideal representation of the mean phenological dynamic throughout the year for each pixel. Therefore, in order to quantify the effects of drought events on the present vegetation, the deviation ΔHarmonics of the original time-series values to the fitted curve were analyzed by subtracting the fitted values from the original ones. Lastly, a temporal subset using the months of May to October was created, since the focus of this study lies on drought analysis during the summer months.

$$X_t = \beta_0 + \beta_1 \cdot 2\pi t + \sum_{i=1}^{n}(\beta_{2i} \cdot cos(\frac{i2\pi t}{T}) + \beta_{3i} \cdot sin(\frac{i2\pi t}{T})) \tag{10}$$

where:

| | |
|---|---|
| $t$ | = Time difference in fractal years since 1 January 1970 |
| $T$ | = Length of time period (one year) |
| $n$ | = Total number of harmonics |
| $i$ | = Current harmonic |
| $\beta_x$ | = Independent coefficients derived from OLS regression |

### 2.3.2. z-Score Analysis

The z-score computation provides a means of data standardization by subtracting the average value of a data set from a single value and subsequently dividing the difference value by the standard deviation of the same data set (Formula (11)) [96]. While many studies commonly use the arithmetic mean as an average value for the z-score calculation [85,97], the median was chosen in this study as it is more robust to outliers compared to the arithmetic mean [98]. Since the phenological characteristics of vegetation vary between different months, a z-score computation for every month within a time-series data set was performed. Identical to the harmonic analysis, the final z-score time-series values were subsetted to only consist of data for the months May until October.

$$z_{ij} = \frac{x_{ij} - \bar{x}_{j-com}}{s_{j-com}} \tag{11}$$

where:

| | |
|---|---|
| $x$ | = Current value |
| $i$ | = Current year |
| $j$ | = Current month |
| $\bar{x}_{j-com}$ | = Median of the current month across all years |
| $s_{j-com}$ | = Standard deviation of the current month across all years |

### 2.4. Analysis Level 1: Best Predictor Combination

As previously mentioned, the first level of analysis was dedicated to identifying the best predictor combination for analyzing vegetation response to drought on a national scale. For this purpose, only MODIS data was applied. 1000 random points across all German forest areas based on the CLC forest classification were hereby sampled and used to extract the values from satellite and meteorological time-series data. It was ensured that only one sample point per pixel was applied. Afterwards, correlation analyses were conducted for each sample point by using the Pearson method.

All data were re-projected to the Universal Transverse Mercator (UTM) zone 32 north prior to any correlation testing. Furthermore, meteorological data was subsetted and resampled to match the temporal coverage and spatial resolution of MODIS data. Consequently, pixel values since 2000 until 2019 and with a 500 m spatial resolution were compared. Moreover, as already mentioned in Sections 2.3.1 and 2.3.2, only the months May–October were considered for correlation testing because the focus of this study lies on the quantification of vegetation response to drought events during summer months.

### 2.4.1. Harmonic Degree Comparison

The first step towards vegetation response analyses was to identify the ideal degree of harmonics needed for modeling the best possible and therefore most representative phenological curve throughout the year. In order to test the suitability of different harmonic degrees, correlation analyses between MODIS NDVI ΔHarmonic and z-score values were conducted. Next to z-score values, correlation analysis was also applied between ΔHarmonic and scPDSI. Pixel values were extracted using the mentioned 1000 sample points across German forests. It was expected that, while not being identical, stronger similarities between a given harmonic degree and z-score or scPDSI values can be interpreted as a better representation of the actual vegetation response.

### 2.4.2. Comparison between Meteorological Drought and Spectral Characteristics of Forests

After the identification of the most suitable harmonic degree, ΔHarmonic and z-score values based on MODIS data were correlated against meteorological data, including scPDSI, maximum air temperature and the sum of precipitation per month. In addition, monthly soil moisture values were compared with remote sensing predictors. All three indices NDVI, NBR and NDMI were hereby investigated in order to identify the most potent vegetation index for drought assessment. Again, the 1000 sample points across German forests were used for correlation testing. Therefore, while a deviation between the vegetational and meteorological analysis is to be expected, the index with the overall highest correlation to meteorological predictors was identified to be most suitable for studying the vegetation response to drought.

Correlation tests were conducted on a monthly and annual basis. For the annual analysis, the annual minimum value was applied in the case of the satellite data (NDVI, NBR, NDMI) as well as for the meteorological predictors scPDSI, precipitation and soil moisture. For maximum air temperature the annual maximum value was used. Since the vegetational response to drought events is analyzed, values for the three indices NDVI, NBR and NDMI as well as for the meteorological predictors scPDSI, precipitation and soil moisture are expected to be low during drought conditions, while maximum temperature is expected to be high during drought events in summer. Additionally, a temporal shift of 1–3 months and 1–3 years, respectively, was included to analyze a potential time lag between the meteorological drought and the vegetation response.

Subsequently, monthly ΔHarmonic and z-score based maps of the vegetation reaction to drought for all German forests were generated using the best performing vegetation index and a harmonic degree in combination with MODIS remote sensing data.

### 2.5. Analysis Level 2: Forest Type Vulnerability

The second analysis level was dedicated to studying drought impact on different forest compositions. This includes the local dominant tree type, forest age as well as the present soil type and topographic parameters. The forest heatmap provided by the BaySF was hereby used as a mask. Only Landsat data was used for Analysis Level 2 since it provides the spatial resolution necessary to conduct detailed local investigations with such small-scale landscape heterogeneity. The most potent vegetation index and harmonic degree, as defined in Sections 2.4.1 and 2.4.2, were utilized. Satellite imagery from August 2018 was used for drought analyses because the summer of 2018 featured heavy drought conditions and was therefore selected as a representative example.

### 2.5.1. Tree Species Vulnerability

In order to compare the vulnerability of different tree species to drought events, z-score and ΔHarmonic values were extracted for each dominant tree species based on the forest reference data set provided by the BaySF. Since the reference data also included information about forest age (older/younger 80 years), the impact of forest age on drought susceptibility was analyzed, as well.

### 2.5.2. Impact of Soil Types on Drought Vulnerability

For studying the impact of soils on drought susceptibility, only areas with the most vulnerable tree species as the local dominant tree type were used to mask the soil data. Therefore, the influence of present tree species compositions could be reduced. In addition, the impact of soils was also analyzed for the whole forest heatmap area. Soil classes with the lowest z-score and ΔHarmonic values were hereby interpreted to have the strongest impact on the drought susceptibility of forests.

### 2.5.3. Topographic Influence on Drought Vulnerability

The last step for the second analysis level was to study the influence of the topography on the drought response of forest areas. Information on the slope was hereby correlated

against remote sensing predictors. Furthermore, the effect of aspect was analyzed by comparing z-score and ΔHarmonic values for north-, east-, south- and west-facing forests. Since the calculation of slope and aspect is based on a DTM with a spatial resolution of 25 m, all data was re-projected to match Landsat data by using a bilinear interpolation approach. The soil types with the most negative influence on drought susceptibility as described in Section 2.5.2 were thereby used as a mask. Thus, the influence of different soil types and tree species compositions on drought vulnerability could be reduced to a minimum. Analogous to the analysis on soil types, the influence of topographic parameters were also tested for the whole forest heatmap area.

## 3. Results

### 3.1. Analysis Level 1: Best Predictor Combination

#### 3.1.1. Harmonic Degree Comparison

Time-series plots with different harmonic fitted curves are displayed in Figure 3. Figure 4a illustrates the spread of correlation scores between MODIS ΔHarmonic and z-score NDVI values. Increasing the harmonic degree resulted in overall higher correlation values with a median $r$ of 0.73 for the 1st harmonic degree, a median $r$ of 0.84 for the 3rd and a median $r$ of 0.93 for the 6th harmonic degree.

Correlation scores between ΔHarmonic and scPDSI values, as seen in Figure 4b, also suggest higher overall correlation values for higher harmonic degrees. Median $r$ scores for the 1st 3rd and 6th harmonic are 0.17, 0.2 and 0.21. Overall $r$ scores are significantly lower ranges are wider compared to the z-score correlation.

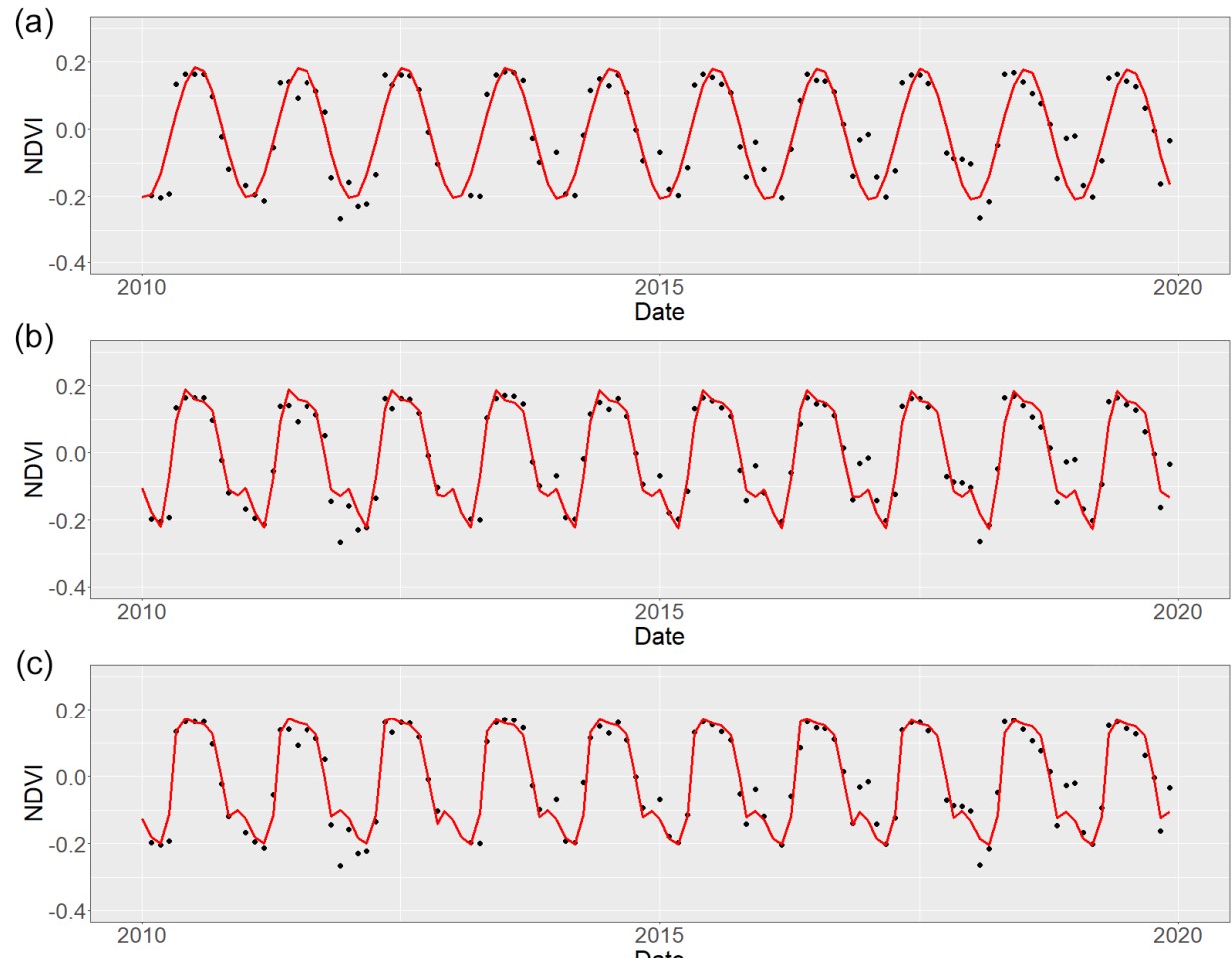

**Figure 3.** *Cont.*

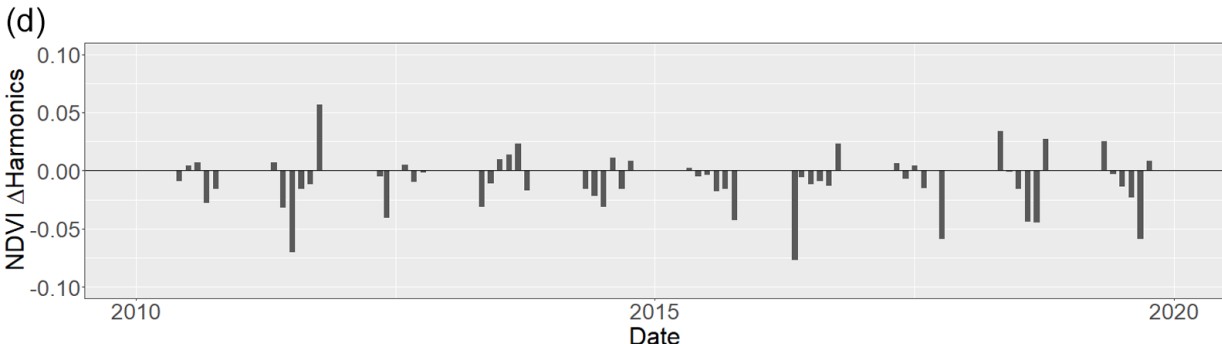

**Figure 3.** ΔHarmonics calculation based on detrended monthly median MODIS NDVI data of the Steigerwald between 2010 and 2019. (**a**) Fitted curve using a 1st harmonic degree. (**b**) Fitted curve using a 3rd harmonic degree. (**c**) Fitted curve using a 6th harmonic degree. (**d**) Difference (ΔHarmonics) between NDVI values and the 6th degree harmonic fitted curve for months May–October.

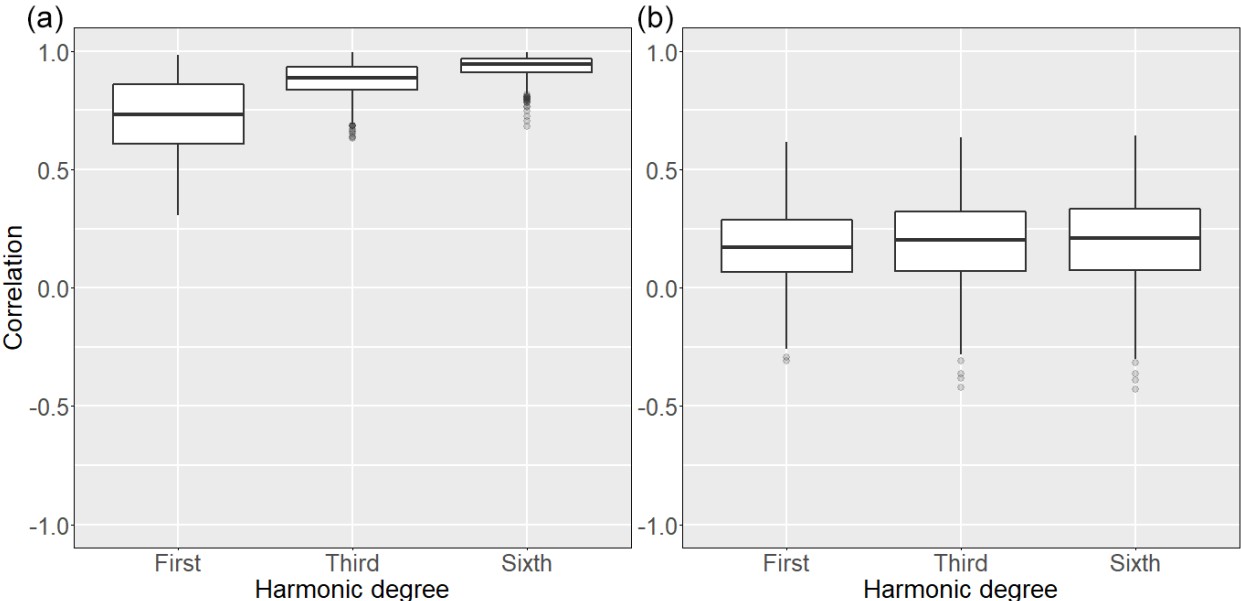

**Figure 4.** Boxplots of correlation scores between monthly MODIS NDVI ΔHarmonic values and MODIS NDVI z-score (**a**) as well as scPDSI (**b**) values. A higher harmonic degree leads to overall higher correlation scores. scPDSI vs. ΔHarmonic *r* values (**b**) are significantly lower compared to z-score vs. ΔHarmonic *r* values (**a**).

### 3.1.2. Comparison between Meteorological Drought and Spectral Characteristics of Forests

Monthly correlation analysis between scPDSI and remote sensing predictors indicated highest *r* scores by using the NDVI (Figure 5). Said index performed best for both ΔHarmonic and z-score approaches. Introducing temporal shifts did hereby not lead to any significant improvements in correlation scores. Highest median *r* of 0.27 for the z-score (Figure 5b) and 0.21 for the ΔHarmonic approach (Figure 5a) were generated with no temporal offset and NDVI as the vegetation index. Correlation numbers for precipitation and soil moisture were overall similar but lower compared to the scPDSI. In contrast, correlation scores for monthly maximum air temperature were for the most part close to 0, regardless of method, index and temporal offset.

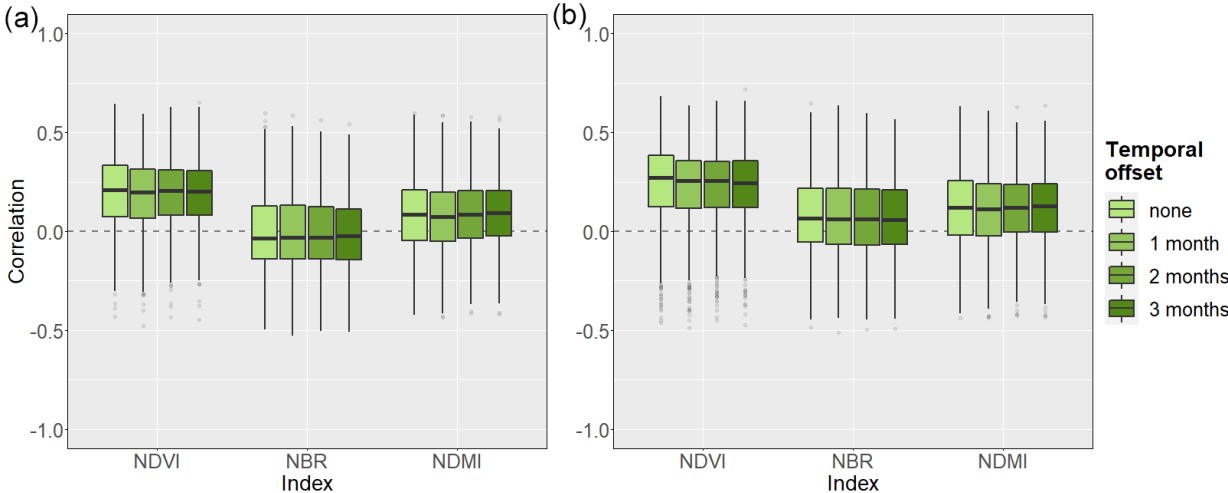

**Figure 5.** Boxplots of correlation scores between monthly scPDSI values and MODIS ΔHarmonic (**a**) as well as z-score (**b**) values. NDVI outperformed the other two indices for both ΔHarmonics (**a**) and z-score (**b**) data. z-score generated overall slightly higher *r* values.

Similar behavior was also observed for annual correlation analysis. In general, highest correlation scores could be generated for the scPDSI (Figure 6), followed by soil moisture. For all meteorological predictors, using the NDVI resulted in the strongest correlations. In case of NDVI and NBR a temporal shift of three years achieved higher *r* scores compared to a one or two-year shift while at the same time the overall strongest correlation was still observed for NDVI without a temporal offset. Since choosing the NDVI led to the highest correlation values in the majority of scenarios, it was therefore considered to be the most potent vegetation index for vegetation analyses in the context of drought response.

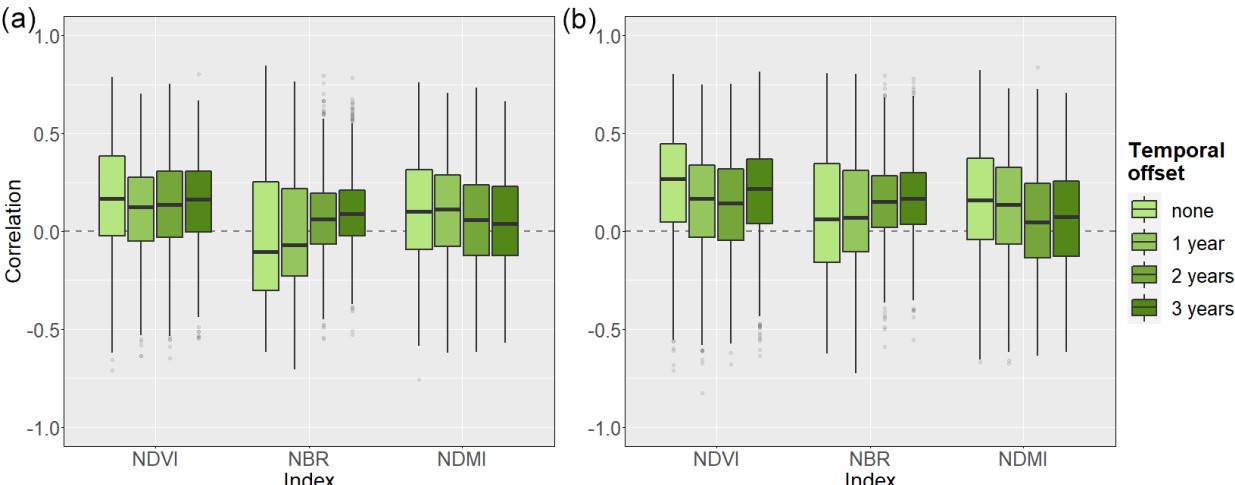

**Figure 6.** Boxplots of correlation scores between annual minimum scPDSI values and annual minimum ΔHarmonics (**a**) as well as z-score (**b**) data. NDVI outperformed the other two indices for both ΔHarmonics (**a**) and z-score (**b**) data. z-score generated overall slightly higher *r* values.

Spatial variability in correlation coefficients between scPDSI and MODIS NDVI z-score as well as scPDSI and MODIS NDVI ΔHarmonic data is illustrated in Figure 7. Both vegetation response detection methods feature a high spatial variability of *r* values across different forest areas. Furthermore, the mean correlation coefficients for each main natural unit after Schmithüsen and Meynen [99], who divided Germany into 86 nature units based on climatic characteristics, soil types and topographic parameters, are displayed in Appendix A Figure A3. Overall slightly higher correlation and lower *p*-values for all German forests were achieved by using the z-score approach.

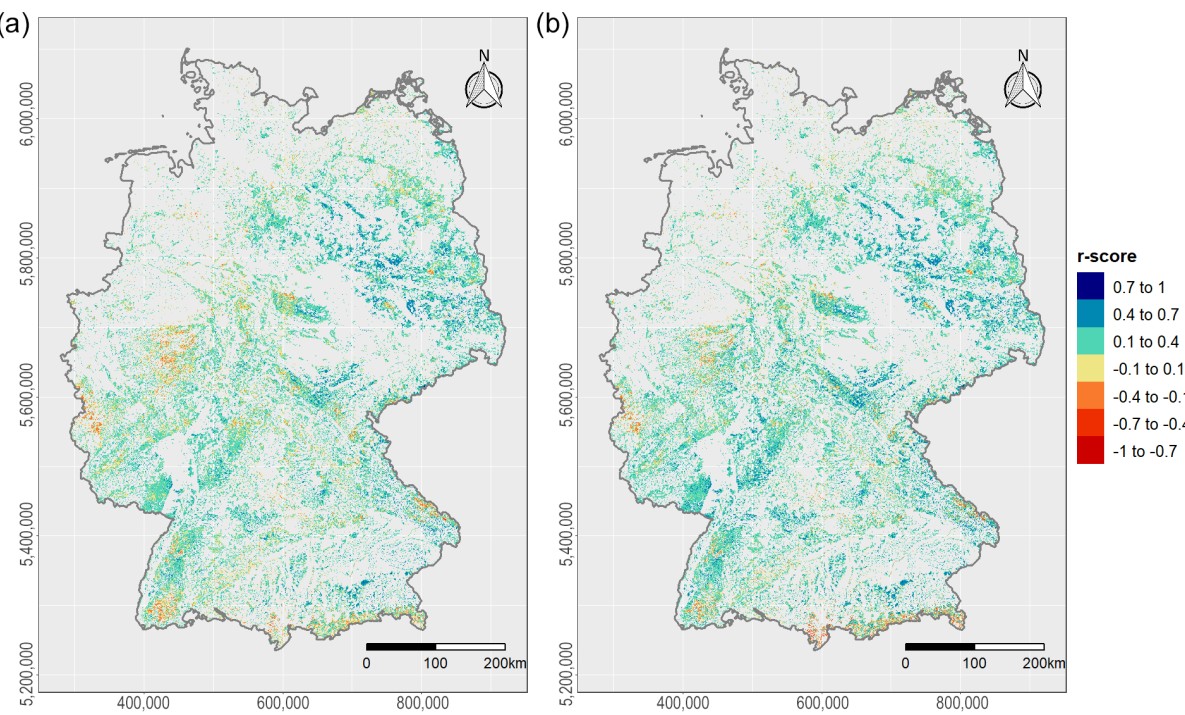

**Figure 7.** Spatial variability of correlation coefficients between scPDSI and (**a**) MODIS NDVI ΔHarmonic as well as (**b**) MODIS NDVI z-score values across German forest areas.

### 3.1.3. Vulnerability Maps of German Forests

Monthly vulnerability maps of German forests were created using the previously identified most potent vegetation index (NDVI) and harmonic degree (6th). MODIS NDVI ΔHarmonic and z-score values together with scPDSI numbers for August 2018 are visualized in Figure 8. All three data sets indicate severe drought conditions across all forests for the mentioned month. Although scPDSI values indicate severe to extreme drought conditions for alpine forests in southern Bavaria, both z-score and ΔHarmonic suggest relatively mild reactions of forests in those regions.

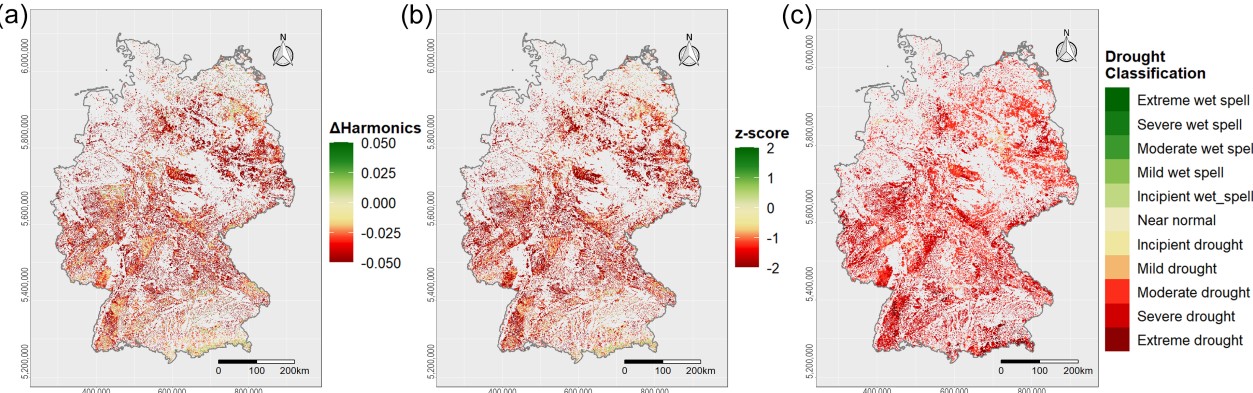

**Figure 8.** Drought maps derived from (**a**) MODIS NDVI ΔHarmonic, (**b**) MODIS NDVI z-score and c) scPDSI values for German forests in August 2018.

### 3.2. *Analysis Level 2: Forest Type Vulnerability*

#### 3.2.1. Tree Species Vulnerability

The distribution of Landsat based NDVI z-score and ΔHarmonic values for each tree type and age in 2018 can be seen in Figure 9. The order of trees is defined by the overall mean value across both forest ages. For both methods, forest areas with pine and larch as

the dominant tree types feature overall lowest z-score and ΔHarmonic values. Overall least negative z-score and ΔHarmonic numbers (close to 0) were generated for forest regions with fir as the dominant tree type.

A comparison between the different forest ages reveals the three most vulnerable tree species (Pine, Larch and Oak) to be generally more heavily affected if forests are older than 80 years. On the other hand, the less heavily affected precious hardwood, spruce and beech-dominated forests feature overall slightly lower z-score and ΔHarmonic values for younger forests (<80 years).

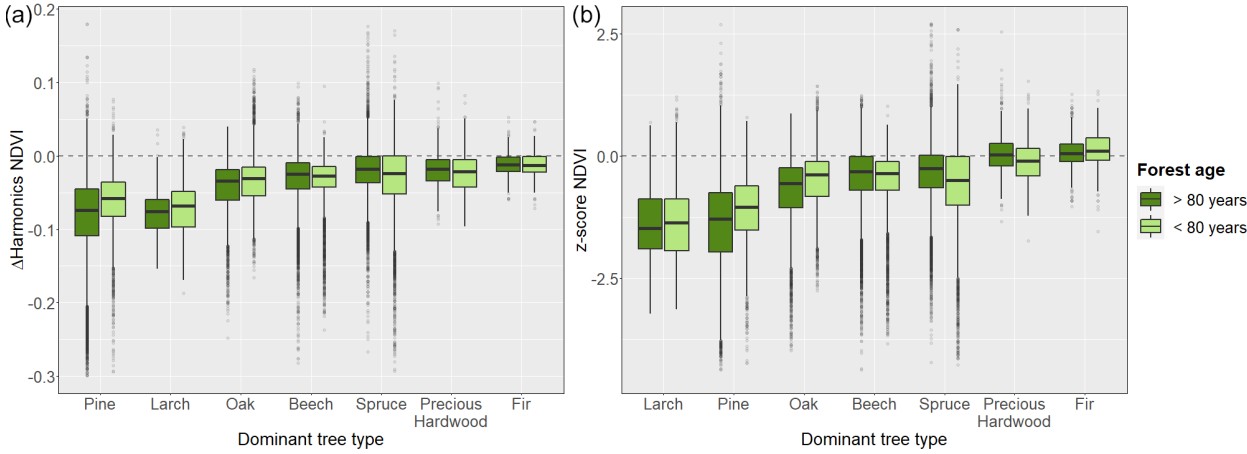

**Figure 9.** Boxplots of (**a**) 6th degree ΔHarmonic NDVI and (**b**) z-score NDVI values in August 2018 for forests with different dominant tree species based on Landsat data. Forests with Larch and Pine as the dominant species feature overall lowest values. Tree types are ordered after the overall mean value.

### 3.2.2. Influence of Soils on Drought Vulnerability

Both pine and larch were identified to be the most heavily affected tree species. Since regions with pine as the dominant tree type cover a larger area (27.3 km²) compared to regions with predominantly larch trees (1.1 km²), only forests dominated by pine trees were used to study the impact of soils.

The five soil types that resulted in both, lowest z-score and ΔHarmonic NDVI numbers are listed in Table 4. The soil types "401a" "529a" and "529b" featured hereby especially low NDVI z-score and ΔHarmonic values compared to other soil classes. The same behavior could also be observed not only for pine stands, but also for the whole forest heatmap area. In contrast, the soil type "407a" demonstrated highest z-score and ΔHarmonic NDVI values. After the "Bodenkundliche Kartieranleitung" (KA5), areas with the soil type "407a" are described to feature almost exclusively brown earth (including pseudogley) over loam or clay (sedimentary rock) with a surface layer of gravel-leading loam [100].

**Table 4.** Soil types which led to lowest z-score and ΔHarmonic NDVI values in August 2018 for forest areas with pine as the dominant tree species. Description for each soil type is based on the KA5 [100].

| Soil | Description |
|------|-------------|
| 401a | Almost exclusively regosol from gravel leading sand to sandy loam, poorly distributed over sandstone. |
| 405a | Predominantly brown earth, poorly distributed pseudogley brown soil, in forest areas poorly distributed podsol made out of (gravel-leading) sand (surface layer or sandstone) over (gravel leading) clay (sedimentary rock). |
| 524b | Almost exclusively brown earth (containing podsol), rarely podsol brown earth from skeletal-leading sand (surface layer) over (skeletal-leading) sand (sandstone). |
| 529a | Predominantly pseudogley, poorly distributed brown earth and podsol pseudogley from gravel leading sand to sandy loam (surface layer, sedimentary rock). |
| 529b | Predominantly pseudogley, poorly distributed brown earth and podsol pseudogley from (skeletal-leading) sand to sandy foam (surface layer, sedimentary rock) over clay (sedimentary rock). |

### 3.2.3. Topographic Influence on Drought Vulnerability

For topographic analyses, only areas that feature pine as the dominant tree type were used. Furthermore, the said area was masked to only cover regions with the previously identified five soils, which led to the lowest z-score and ΔHarmonic values (Table 4). That way, the influence of different dominant tree and soil types were reduced to a minimum for topographic investigations.

Correlation analysis between remote sensing predictors and the topographic variable slope revealed a positive relationship of $r = 0.26$ for z-score NDVI values and $r = 0.32$ for ΔHarmonic NDVI. Similar results were generated for the whole forest heatmap with $r = 0.25$ for z-score NDVI values and $r = 0.31$ for ΔHarmonic NDVI. This indicates less severe reactions of the vegetation to meteorological drought events on slopes compared to flat landscapes.

Lastly, no significant influence of the aspect on either z-score or ΔHarmonic NDVI values could be observed. Satellite-derived values were overall nearly identical for each compass direction.

## 4. Discussion

### 4.1. Spatio-Temporal Availability of Applied Satellite Data

Remote sensing data derived from five different satellites were used within this study. Imagery from Landsat-4, -5, -7, -8 and MODIS Terra was applied. A single continuous Landsat product was generated by fusing data from the mentioned Landsat satellites and calculating monthly median images since 1984. Independent from the two analysis levels, the frequency of available data across Germany from both MODIS and Landsat was compared. For this purpose, Landsat imagery was resampled to match the spatial resolution of MODIS data (500 m) by using a bilinear interpolation approach. Landsat time-series was hereby further subsetted to match the temporal window of MODIS (since February 2000). Thus, the number of available monthly median scenes after cloud, cloud shadow and snow masking from February 2000 until December 2019 could be investigated on a per-pixel-basis.

Different numbers of scenes are available depending on the satellite source. Landsat features both a higher spatial resolution and a longer temporal coverage compared to MODIS. Furthermore, since data from several Landsat missions were combined, the number of available Landsat images vary across time. A general increase in the number of cloud-, cloud shadow- and snow-free Landsat scenes since 1984 can be observed (Figure 10). With the launch of Landsat-5 in 1999, imagery from two Landsat satellites were available at the same time from that point on forward. This resulted in an overall increase in available data. However, between the last available SR data from Landsat-5 in May 2012 and the start of Landsat-8 in February 2013, imagery from only one satellite, Landsat-7, was available [42,43]. Moreover, the Scan Line Corrector (SLC) of the Landsat-7 Enhanced Thematic Mapper Plus (ETM+) sensor failed on 31 May 2003 [101]. This led to wedge-shaped scan gaps within the satellite imagery and therefore reduced the amount of available data. Calculating monthly median images should largely account for this issue but effectively there was still fewer data available for this time span.

In contrast to Landsat data, MODIS imagery features a higher temporal frequency and a more consistent spatial coverage (Figures 11 and 12), but at the cost of a lower spatial resolution. A higher number of available images per month increases the chances of having one or more cloud-free observations and therefore having more robust and representative median calculations. Landsat data features lower data continuity and a higher noise level compared to MODIS which can be attributed to the lower temporal resolution. The available cloud, cloud shadow and snow masks in the "pixel_qa" quality-band of the Landsat SR products are derived from the CFMask algorithm [43]. In a recent study by Foga et al. [102], the performance of different cloud masking algorithms were validated on a total of 96 globally distributed Landsat-8 scenes. The CFMask algorithm featured hereby the best overall accuracy results for both cloud detection (roughly 90%)

and cloud shadow detection (roughly 96%) [102]. Known issues of the CFMask are the detection of clouds over bright surfaces, the detection of clouds for areas with temperature differentials that are either too small or too large between the clouds and the surrounding surface as well as the detection of optically thin clouds [43]. Despite these known issues, which could potentially impair the quality of the Landsat time-series, we consider the use of the cloud, cloud shadow and snow masks generated by the CFMask algorithm to be a reasonable choice as recommended by Foga et al. [102].

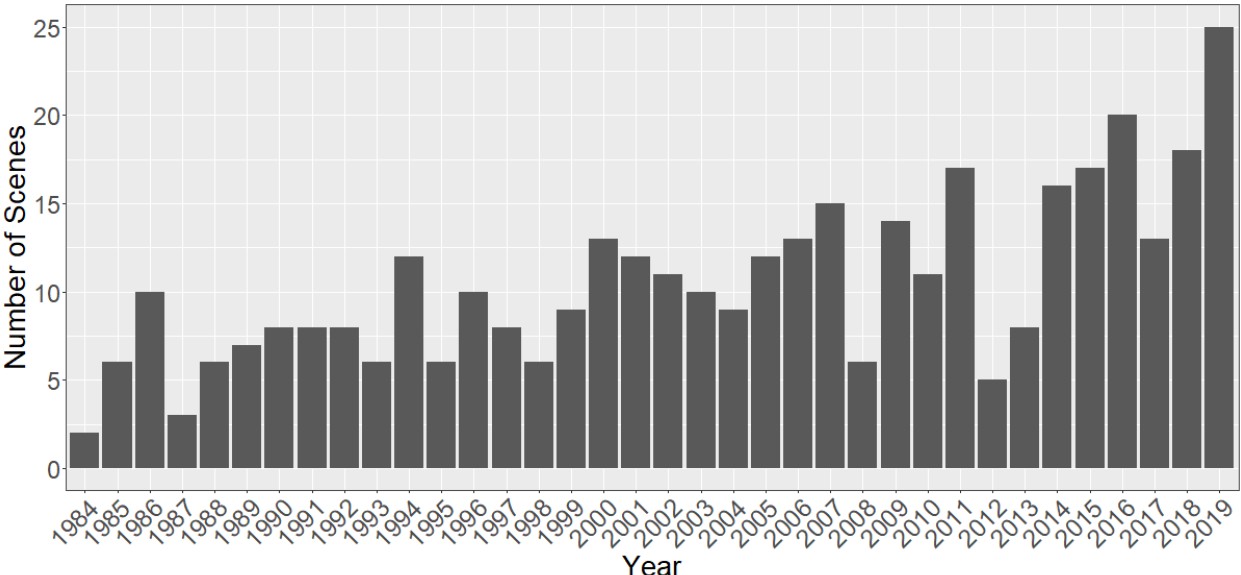

**Figure 10.** Frequency of combined Landsat data per year from the sensors TM, ETM+ and OLI of the Steigerwald. The remote sensing imagery was masked from clouds, cloud shadows and snow. An overall increase in data frequency can be observed.

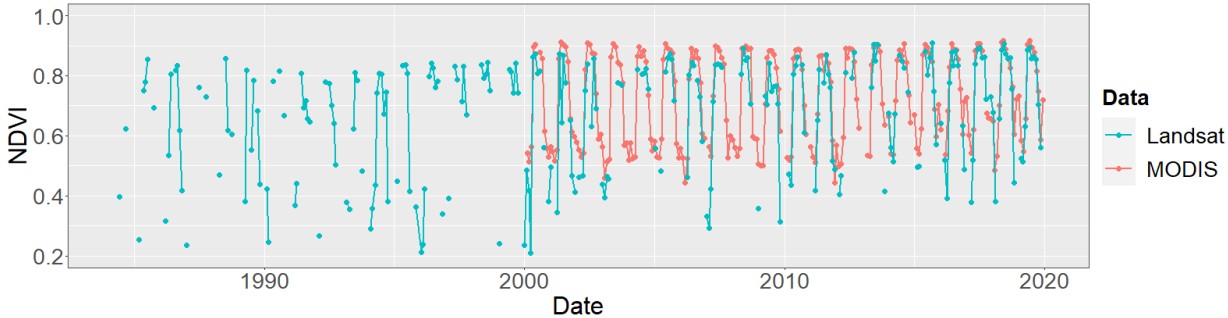

**Figure 11.** Monthly median NDVI values for Landsat and MODIS data of the Steigerwald. Remote sensing imagery was masked from clouds, cloud shadows and snow. Adjacent months with available data are connected with a line. MODIS data has a smaller temporal coverage compared to Landsat data, but provides less noisy values. Furthermore, MODIS features a higher data continuity than Landsat.

Figure 12 illustrates the number of months with available imagery since the year 2000 across Germany. MODIS data features an overall higher and more homogeneous data frequency (Figure 12c). Contrary to that, Landsat imagery shows distinct stripes across Germany with lower data availability (Figure 12b). Areas of low data frequency overlap with the "gaps" of the Worldwide Reference System (WRS) tiles due to the positioning of the Landsat orbit footprints (Figure 12a). As a result of the northward increasing overlap of Landsat orbits, a more consistent data availability in northern Germany can be observed compared to southern Germany.

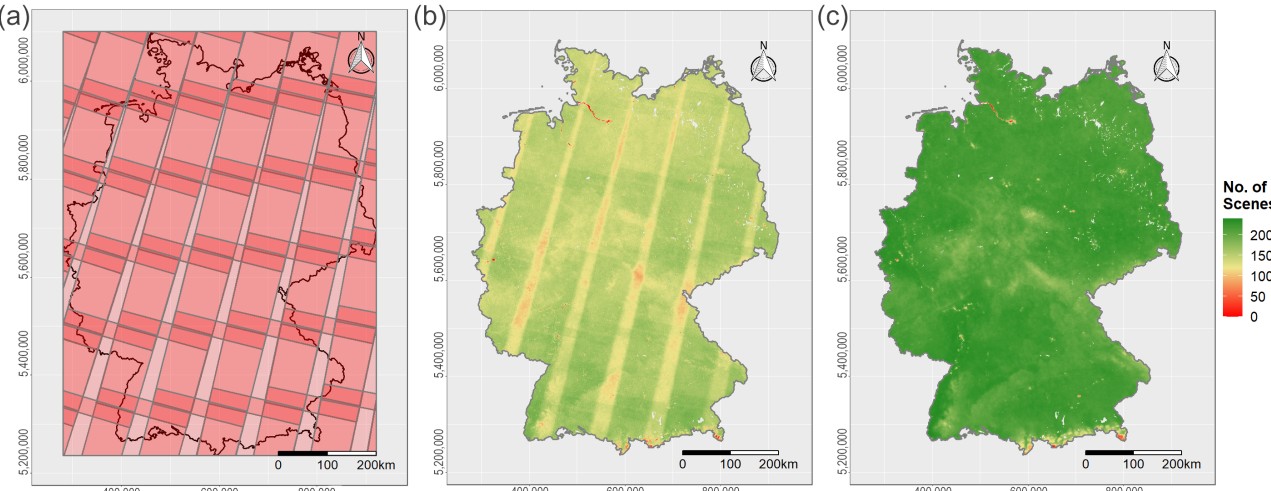

**Figure 12.** Number of monthly median images after cloud, cloud shadow and snow masking since the year 2000. Landsat imagery (**b**) exhibits inconsistent data frequency across Germany, while MODIS data (**c**) features overall more homogeneous and higher data frequency. Areas with lower numbers of available Landsat scenes overlap with tile gaps of the Worldwide Reference System (WRS) as seen in (**a**).

As seen in Table 3 and Appendix A Figure A2, bands of the different sensors Thematic Mapper (TM) (Landsat-4 and -5), Enhanced Thematic Mapper Plus (ETM+) (Landsat-7) and Operational Land Imager (OLI) (Landsat-8) slightly deviate in their covered wavelength areas. It also has to be mentioned that different SR processing algorithms are applied for different Landsat products. Namely, the Landsat Ecosystem Disturbance Adaptive Processing System (LEDAPS) algorithm is used for Landsat-4, -5 and -7, whereas the Land Surface Reflectance Code (LaSRC) is applied for Landsat-8 imagery [42,43]. Both algorithms produce similar results, but utilize different methods for generating SR products [103]. Furthermore, spectral bands of MODIS imagery also feature differences in spectral wavelength coverage compared to Landsat sensors (Table 3). Time-series analyses within this study were based on the normalized indices NDVI, NBR and NDMI. Therefore, differences in spectral coverage are expected to be averaged out for the most part.

With the launch of the Landsat-8 satellite in 2013, earlier studies dedicated their research effort to the comparison between NDVI values generated by different Landsat sensors. Results of these studies show slightly higher NDVI values for Landsat-8 data compared to Landsat-7 imagery, while a high degree of similarity between the two sensors was stated especially for areas of high vegetation density, such as forests [104,105]. Within this context, Roy et al. [106] suggested a simple linear OLS regression equation for both Top-of-Atmosphere (TOA) and SR data, in order to convert Landsat-8 NDVI products to match Landsat-7 and vice versa [106]. While the quality of time-series investigation could have potentially increased by introducing a regression equation, no such regression was applied during this study since only areas with a high vegetation density (forests) were analyzed for which the difference in NDVI was found to be close to 0 [105]. In another study done by Zhang and Roy [107], inconsistencies within a 27-year NDVI time-series using Landsat-5 data were examined. Linear regression analysis revealed a minor negative trend due to changes in orbit positioning [107]. No correction to these inconsistencies was performed on Landsat-5 data within this study which might have further increased the comparability between different Landsat data sets.

*4.2. Analysis Level 1: Best Predictor Combination*

ΔHarmonic NDVI values derived from MODIS data were correlated against scPDSI and z-score values using the 1st, 3rd and 6th harmonic degree. As seen in Figure 4, increasing the harmonic degree resulted in overall higher correlation scores, with the 6th degree of harmonic performing best out of all tested degrees. Similar results were

demonstrated in a study by Wilson et al. [33], who correlated a harmonic time-series using the 1st to 4th degree against national forest inventory data. The root-mean-square error (RMSE) hereby decreased with an increasing harmonic degree [33]. Improvements in correlation scores were greater between the 1st and 3rd degree, compared to the 3rd and 6th degree. This mirrors results of other studies which suggest 90 percent of variations in time-series of vegetation phenology is explained by the first three harmonic degrees [34,108].

Figure 4 further demonstrates significantly higher $r$ scores between z-score and ΔHarmonic values compared to correlation scores between ΔHarmonic and scPDSI. This exemplifies that the vulnerability of a forest is not only dependent on meteorological conditions but, as also mentioned by Sykes [1], on other additional biotic and abiotic factors such as water availability, available nutrients, forest composition, forest age, topographic positioning or pest infestation. That said, scPDSI data generated the overall highest correlation scores out of all meteorological predictors, followed by soil moisture. Maximum air temperature featured the lowest $r$ values. Similar findings were observed in a study by Scharnweber et al. [62], who used tree ring data to estimate past climatic variability and correlated the tree ring data against meteorological values in north-eastern Germany. The highest correlation scores were hereby achieved by using the scPDSI, followed by soil moisture and precipitation, whereas temperature featured the overall weakest correlation [62]. Another study by Büntgen et al. [61] compared tree ring information of oak trees in central Germany with meteorological drought. scPDSI outperformed temperature data, for which no significant correlations were found [61]. While some studies demonstrated good correlation results between remote sensing and temperature data in other climatic regions (e.g., [109,110]), results derived from previous studies related to German forests as well as results generated within this study suggest both soil moisture and in particular scPDSI to be among the most potent meteorological drought predictors for similar climatic regions.

The vegetation index NDVI performed best in most scenarios and was therefore recognized as the most potent index for quantifying vegetation response to drought events within the framework of this study. Although the NDMI is considered to be a good indicator for vegetation moisture content and also featured significant correlation scores to meteorological predictors, it was not able to surpass the performance of the NDVI. While in a previous study by Gu et al. [111] the NDMI values exhibited a quicker response to drought conditions than NDVI in grasslands, our results suggest that using the NDVI, which reflects the photosynthetic activity and vitality of a plant by including the red band in addition to the NIR band [71], is a better proxy for assessing the vegetation response to drought events for forest canopies.

Investigations related to temporal offsets between meteorological conditions and vegetational response revealed generally the strongest correlation without any offset. That said, for annual imagery a temporal offset of three years resulted in slightly higher overall $r$ scores compared to a one year offset for NDVI and NBR correlated against scPDSI (Figure 6), while highest correlation was still generated for NDVI without any time lag. The two previously mentioned studies by Büntgen et al. [61] and Scharnweber et al. [62] also identified worse correlation results for a one-year offset. In contrast, another study by Chuai et al. [112] dedicated their analysis to the relationships between NDVI, precipitation and temperature in Inner Mongolia, China. Significant time-lag effects between the spring and summer season were hereby identified for precipitation, but no effects could be observed for temperature [112]. Using longer temporal shifts as the ones applied in this study could potentially lead to higher correlation results.

A high spatial variability in correlation values between meteorological drought and the response of vegetation could be observed (Figure 7). Other studies also described strong variations in $r$ scores between remote sensing and meteorological data across different climatic districts and vegetation types [112,113]. Such correlation maps provide valuable information on the degree of relationship between meteorological and vegetation data across space. Therefore, areas that are more dependable on climatic conditions can be distinguished from areas that are less sensitive to extreme meteorological events,

e.g., drought. In general, a significant relationship between meteorological conditions and vegetation response could be observed. However, the varying degree of correlation strength demonstrates that the health status of a forest is not only dependent on the current meteorological condition, but, as previously mentioned, on a combination of various biotic and abiotic factors.

Overall higher correlation scores could be achieved by using the z-score method when comparing meteorological drought with vegetation response. However, correlation analysis between remote sensing data and the topographical predictor slope suggests higher performance for the ΔHarmonic approach. While both methods generated results that are strongly correlated to each other, distinct differences in the results of the two algorithms could be observed throughout this study. Quality assessment for the two presented methods (z-score and ΔHarmonics) as well as for the different remote sensing data sets (MODIS and Landsat) is limited due to the absence of in-situ drought quantification data. As mentioned by Pause et al. [6] and Lausch et al. [21], in-situ sampling data is required for adding value to physical observations based on satellite data and for interlinking forest health with abiotic and biotic parameters. That said, the authors further mention that in-situ observations vary in type, quality and quantity, which in turn hinders the exploitation for satellite-based analyses. A standardization of drought-related ground truth data is, therefore, a key aspect for improving our understanding of the connection between spectral response and forest health status [6,21].

*4.3. Analysis Level 2: Forest Type Vulnerability*

Studying the response of different tree types to drought conditions in August 2018 revealed the weakest susceptibility to drought for fir, spruce and beech trees (Figure 9). Both, fir and beech trees are often considered to be resistant and resilient future tree types in the face of climate change-induced extreme weather events [4]. Spruce, on the other hand, as a shallow-rooted tree, is commonly described to be strongly prone to drought and windfalls [114]. Furthermore, spruce trees in many forests are strongly affected by bark beetle infestations which led to large-scale felling of infected trees [114,115]. No information about pest infestation was included within the framework of this study which might have increased the comparability of different tree species in their drought vulnerability. It must also be stressed that the included reference data of dominant tree types is limited in its spatial extent by only having information about selected Bavarian forest regions (Figure 2). Results for German-wide analyses might differ due to varying local and regional climatic conditions and soil properties, as suggested by the main natural units after Schmithüsen and Meynen [99]). On the other hand, the strongest negative response to drought events were observed for forest areas with pine or larch as the dominant tree type. Large-scale dying of pine stands after long droughts in German forests matches these results [14]. Pine forests are further troubled by fungal infestations of *Sphaeropsis sapinea*, causing dieback of affected trees [116,117]. As already mentioned, no information about pest infestation was included for the user-reference data. Larch trees were also identified to be among the most vulnerable tree types. While the European larch is widely used in intensive breeding programs due to its fast growth rate and high wood quality [118], recent studies identified a high vulnerability to low soil moisture contents and drought events, especially in mountainous and alpine regions [119–121].

Somewhat divided results in vulnerability were observed for different forest ages depending on the dominant tree type. On the one hand, the generally more susceptible tree types pine, larch and oak demonstrated a general higher vulnerability to drought in forest stands which are older than 80 years. On the other hand, the impact of forest age on less vulnerable tree types proved to be weaker. In a study by Bottero et al. [122], the vulnerability of forest ecosystems depending on, amongst other factors, stand age was studied. Drought was hereby independent of the age of a forest [122]. Drought vulnerability also increases the risk of forest fires [20]. In case of boreal forest regions, the probability of burning events is generally assumed to be independent of forest age [123].

However, some studies suggest an increase in burning probability until the age 50 [124,125]. That said, only forest ages older or younger than 80 years were compared during this study. A lower age threshold might thereby lead to more distinct differences.

It also has to be mentioned that only the dominant tree type of a given forest area was provided. No information about other tree types and their proportions in a given area was available. Lastly, since only the dominant tree type of an area is known, no information about intra-specific variations within one tree type were studied.

The impact of different soils was studied using only one dominant tree type. Thus, the influence of various tree species on soil-dependent drought vulnerability analysis was reduced to a minimum. Pine was hereby selected since it proved to be one of the most responsive trees to drought conditions (Figure 9) and at the same time covered a fairly large area compared to other sensitive tree types within the reference data set. Results revealed all five soil classes which led to the lowest z-score and ΔHarmonic values to feature sandy components (Table 4). The same results could also be observed for the whole forest heatmap area. In contrast, soil type "407a" led to least negative z-score and ΔHarmonic values. This soil type differs from the soils mentioned in Table 4 by having a higher amount of brown earth and no explicitly described sandy components [100]. In the previously mentioned study by Reif et al. [4], forests on sandy soils were also described to be heavily endangered. Pine trees are generally considered to be undemanding in terms of soil quality and are therefore able to grow on e.g., sandy locations better compared to other competing tree types [126]. However, in terms of drought conditions, different soil types can have a significant influence on the vulnerability of pine trees as seen in this study.

Correlation analysis between remote sensing predictors and the topographic parameter slope revealed a positive correlation which suggests less severe drought response on slopes. A possible reason for this behavior could be the potential redistribution of soil moisture content along a hillslope gradient. In a recent study by Hawthorne and Miniat [127], higher soil moisture content at the cove area of a plot was observed. Hence, topography might have mitigated drought effects especially in lower slope positions [127]. That said, no attention was given to the slope position along a gradient, but only the degree of the slope was considered within this study. Moreover, the majority of pixels feature slopes of 15° or less. Further analyses on steeper slopes and with respect to the position on a slope gradient are recommended for future studies.

## 5. Conclusions

Within this study, the potential of harmonic modeling and z-score standardization for detecting and quantifying vegetation response of forest ecosystems to drought events was analyzed via satellite remote sensing data. Both approaches generated results that were strongly correlated to each other, but featured slight differences in their performance when correlating against meteorological and topographical data. While the overall performance of the z-score was higher when comparing to meteorological predictors, ΔHarmonic values showed stronger relationships to topographical data. Out of all meteorological data, soil moisture and in particular the self-calibrated Palmer Drought Severity Index (scPDSI) demonstrated the strongest correlations, with maximum air temperature data featuring the weakest relationships to vegetation response. Moderate Resolution Imaging Spectroradiometer (MODIS) imagery provided more consistent and less noisy data in comparison to Landsat, but is limited by both its temporal coverage and its relatively low spatial resolution. Out of the three vegetation indices used, the Normalized Difference Vegetation Index (NDVI) led to overall best results in terms of drought-related spectral response. Pine and larch trees were identified to feature the strongest negative response to drought events, especially pine trees on sandy soils. Lower drought vulnerability was observed for higher slope degrees, potentially caused by a redistribution of the soil moisture content. Both the z-score and ΔHarmonic modeling using a 6th harmonic degree in combination with monthly MODIS NDVI data can be recommended for similar studies on large scales. In particular, the generated maps which display the correlation strength

between satellite-derived vegetation indices and meteorological data provide valuable information on the vulnerability of vegetation to extreme weather conditions across space. The proposed methods and data allow for a near real-time observation of vegetation response of German forests to extreme weather events, such as drought, and can, therefore, be used as a tool for risk reduction. Nevertheless, for further evaluation and detailed quality assessment of the presented methods and data, standardized, nationwide and drought-related in situ measurements are required.

**Author Contributions:** M.P. conceptualized the study design, processed, analyzed, and visualized the data, and wrote the original manuscript. M.W. and C.K.-F. contributed to the study concept, the writing, and the editing of the manuscript. All authors have read and agreed to the published version of the manuscript.

**Funding:** This publication was supported by the Open Access Publication Fund of the University of Wuerzburg.

**Institutional Review Board Statement:** Not applicable.

**Informed Consent Statement:** Not applicable.

**Data Availability Statement:** With the exception of the forest reference data provided by the Bayerische Staatsforsten (BaySF), as this data set is not owned by the authors, all other data that support the findings of this study are available from the corresponding author upon reasonable request.

**Acknowledgments:** We would like to express our great appreciation for the Bayerische Staatsforsten (BaySF) and in particular to Kay Müller for providing the forest reference data set across Bavaria. We would further like to thank the four reviewers for their helpful comments and suggestions on this manuscript.

**Conflicts of Interest:** The authors declare no conflict of interest. The funders had no role in the design of the study; in the collection, analyses, or interpretation of data; in the writing of the manuscript, or in the decision to publish the results.

## Acronyms

| | |
|---|---|
| **BKG** | Bundesamt für Kartographie und Geodäsie |
| **BaySF** | Bayerische Staatsforsten |
| **CDC** | Climate Data Center |
| **CLC** | Corine Land Cover |
| **DTM** | Digital Terrain Model |
| **DWD** | Deutscher Wetterdienst |
| **EEC** | existing climatic conditions |
| **ETM+** | Enhanced Thematic Mapper Plus |
| **GEE** | Google Earth Engine |
| **IDW** | inverse distance weighting |
| **LaSRC** | Land Surface Reflectance Code |
| **LEDAPS** | Landsat Ecosystem Disturbance Adaptive Processing System |
| **LFU** | Landesamt für Umwelt |
| **MODIS** | Moderate Resolution Imaging Spectroradiometer |
| **NBR** | Normalized Burn Ratio |
| **NDMI** | Normalized Difference Moisture Index |
| **NDVI** | Normalized Difference Vegetation Index |
| **NDWI** | Normalized Difference Water Index |
| **NIR** | Near Infrared |
| **OLI** | Operational Land Imager |
| **OLS** | Ordinary Least Squares |
| **PDSI** | Palmer Drought Severity Index |

| **RGB** | Red-Green-Blue |
|---|---|
| **scPDSI** | self-calibrated Palmer Drought Severity Index |
| **SLC** | Scan Line Corrector |
| **SR** | Surface Reflectance |
| **SWIR** | Short Wavelength Infrared |
| **TM** | Thematic Mapper |
| **TOA** | Top-of-Atmosphere |
| **UTM** | Universal Transverse Mercator |
| **WRS** | Worldwide Reference System |

## Appendix A

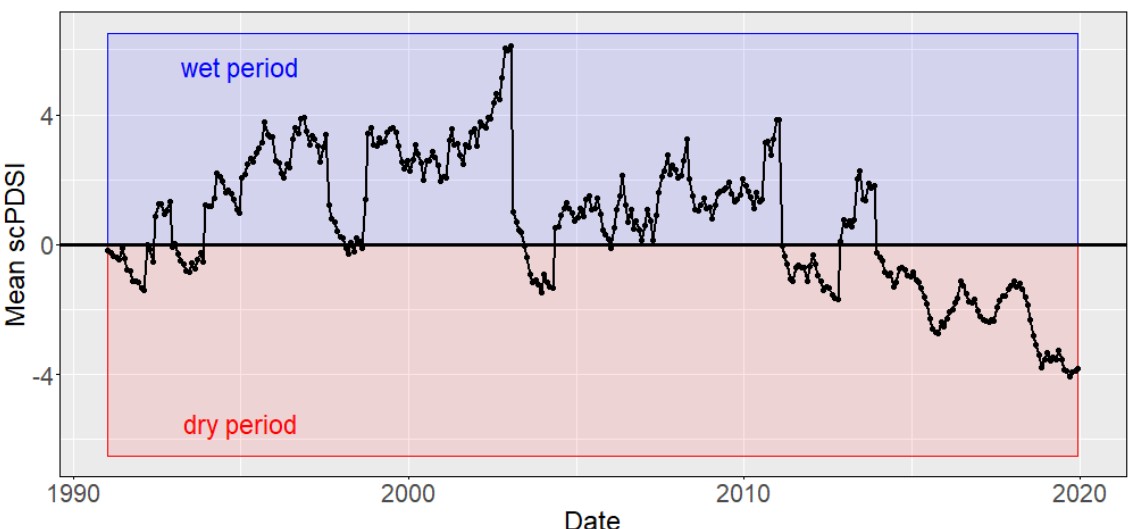

**Figure A1.** Mean scPDSI of the Steigerwald. More recent years feature significantly lower scPDSI values.

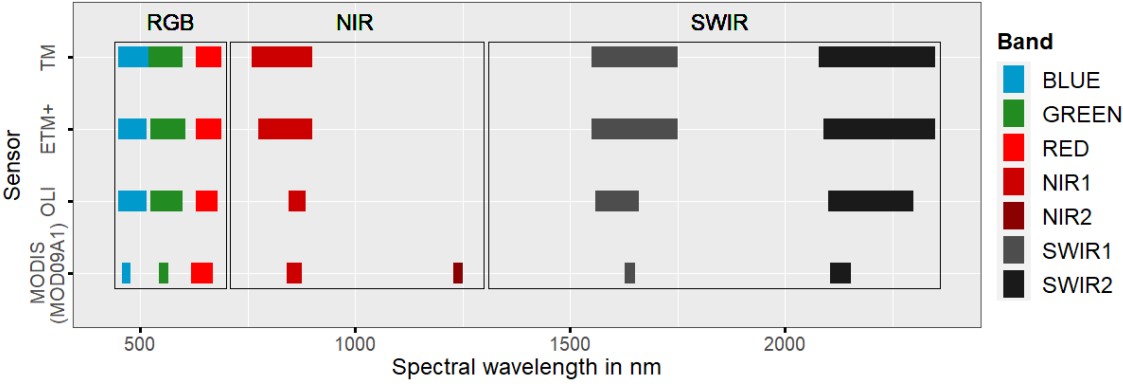

**Figure A2.** Spectral wavelengths covered by different sensors within the visible RGB, NIR and SWIR regions. Bands of the sensors TM [66], ETM+ [67], OLI [67] and the MODIS product MOD09A1 [56] are compared.

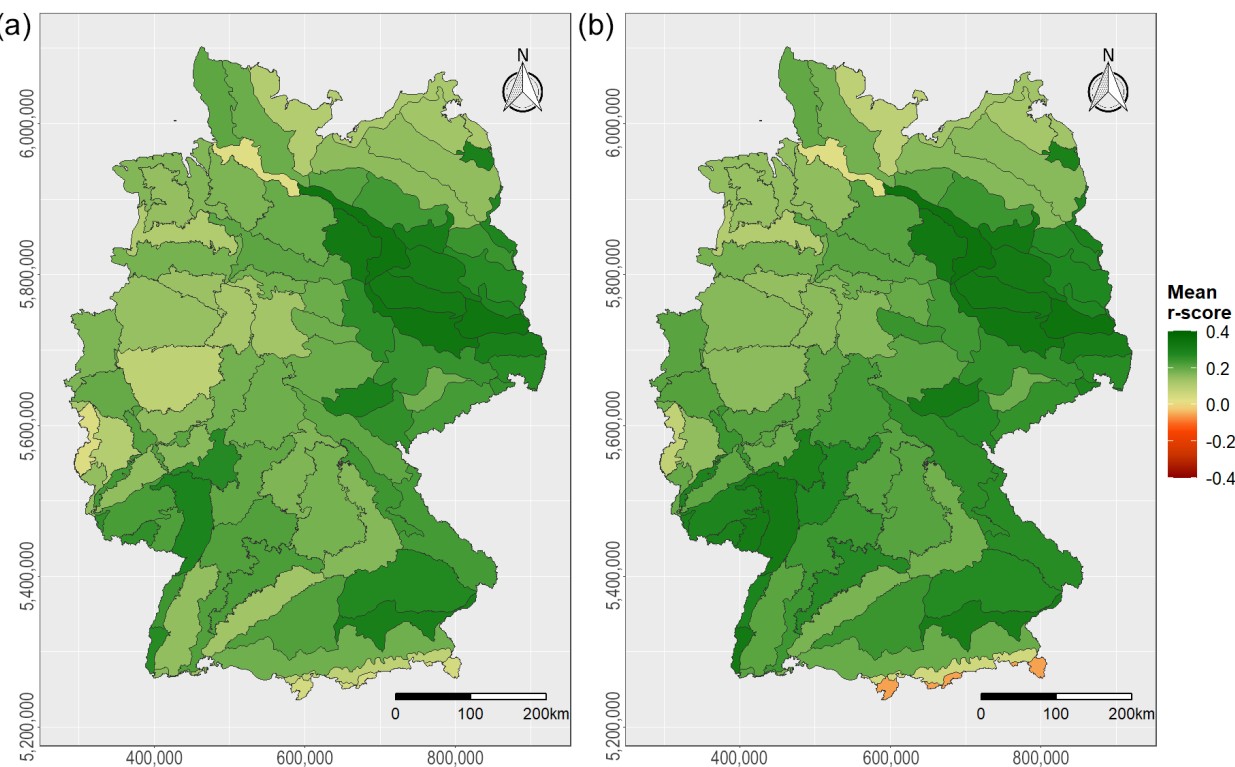

**Figure A3.** Mean correlation coefficients between scPDSI and (**a**) MODIS ΔHarmonic NDVI as well as (**b**) MODIS z-score NDVI values for each main natural unit in Germany after Schmithüsen and Meynen [99], who divided Germany into 86 nature units based on climatic characteristics, soil types and topographic parameters.

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
