# Peer review of "Quantifying the Response of German Forests to Drought Events via Satellite Imagery"

_remotesensing, doi:10.3390/rs13091845_

Round 1

Reviewer 1 Report

See details in the attached file.

Author Response

We thank the reviewer for taking the time to review our manuscript.
Please find our detailed answers to each comment in the attached pdf.

Reviewer 2 Report

Dear all,

Thank you for the opportunity to read this interesting paper.

This work explores the potential of satellite data in combination with harmonic analyses for quantifying the vegetation response to drought events in German forests. This is a very important and interesting approach to monitor the climate change consequences and to reduce risks. From this perspective, I think the study makes an important contribution to the scientific literature and is worthy of publication.

I think the work is well structured and the aim is scientifically useful. The methodology is described in detail, the results presented and discussed clearly and comprehensively.

For these reasons, I believe the paper is already ready for publication. Only the below aspects should be addressed by the authors before publication.

SPECIFIC COMMENTS:

- Section 1.2: “Forest Health in the Face of Climate Change” maybe it’s better “Forest Health during Climate Change”

I think that the formatting of the text should be revised. For instance, Figure 1, 2 are a bit cropped.

Author Response

We thank the reviewer for taking the time to review our manuscript.
Please find our detailed replies to each comment in the attached pdf.

Reviewer 3 Report

The study on effects of drought on forests is a hot topic that monitoring forest health still a challenge under global climate change. Based on Landsat and MODIS imagery, the manuscript used harmonic analysis and z-score method to evaluate vegetation indices response to drought conditions. The framework of the study is complete and descripted in detail. There are several concerns I think should be further clarified.

  1. The topic of the study is about the effects of drought event on forest. In the introduction part, the topic was extended to forest heath. The effects of drought event on forest maybe is one of forest health topic, but not all. The description in introduction need to focus on the effects of drought event on forest ecosystems. In addition, some of the citations in part 1.2 of the introduction are written by German. If possible, could you replace other English references for most people can study easily.

  1. Some figures (e,g, Figure 1-2, Figure 5, Figure 10-12, Figure A14, Figure A15) in the manuscript (for my review version) are not showed in complete size. Some information was missed.

  1. P5 L182 trough --> through

  1. In part 2.2, how to calculate PDSI and scPDSI were still confused. Did you interpolate the meteorological data into gridded pixel? If yes, which method did you use? How to evaluate the interpolation result? Could you give the equation or expression of PDSI to let readers can easily repeat your steps? Similarly, how to calculate scPDSI?

  1. In part 2.5, how many Landsat images did you use? As you mentioned SLC-off in Landsat-7, did you discard these data? Some middle area of Landsat-7 imagery that not affected by SLC-off can be used, did you use these data?

         Line 539 in P19, I also a little confused why you said no regression was           applied could improve the quality of time-series analysis. In the                       previous study (Roy et al., RSE, 2016), Landsat-5, Landsat-7, and                     Landsat-8 data should be matched SR before combined use.

  1. P7 L243, Table 3 should be located in the main body, not in Appendix

  1. P8 L262 Normally, the GEE code link used in the study as supported materials is preferred. This can help readers to repeat your method easily and quickly.

Author Response

(The authors gave the same response as above.)

Reviewer 4 Report

Some comments are given in the paper. If I understand well the correlation between weather condition and NDVI is not so good. I think it is important to assess the impact on this of the potential errors in cloud masks or shadow masks. The drought should impact the moisture in the green vegetation, that theoretically is estimated by the NDMI, then, do you have an explanation why the correlation of the drought with NDMI is lower than with NDVI ?  

Author Response

(The authors gave the same response as above.)

Round 2

Reviewer 1 Report

See Attachment

Author Response

We thank the reviewer for again taking the time to review our manuscript. Please find our detailed answers to each comment/suggestion in the attached PDF.

Reviewer 3 Report

I think the manuscript has been qualified for publication. 

Congratulation! 

Author Response

We want to thank the reviewer again for taking the time to review our manuscript. Thanks to the reviewers helpful comments and suggestions, we were able to greatly improve the quality of our manuscript. We further want to thank him/her for evaluating it to be ready for publication after major revisions.

Reviewer 4 Report

No further comments

Author Response

We want to thank the reviewer again for taking the time to review our manuscript. Thanks to the reviewers helpful comments and suggestions, we were able to greatly improve the quality of our manuscript. We further want to thank him/her for this positive evaluation after major revisions.